# Species-Specific Spillover Patterns Detected by Biomass Gradients in Mediterranean Marine Protected Areas

**Just Tomàs Bayle-Sempere** [1,*], **Aitor Forcada-Almarcha** [2], **Pablo Sánchez-Jerez** [1], **Mireille L. Harmelin-Vivien** [3], **Laurence Le Diréach** [4], **Eric Charbonnel** [5], **José Antonio García-Charton** [6], **Denis Ody** [7], **Olga Reñones** [8], **Carlos Valle** [2] and **Ángel Pérez-Ruzafa** [6]

1   Department of Marine Science and Applied Biology, IMEM "Ramón Margalef", University of Alicante, 03080 Alicante, Spain; psanchez@ua.es
2   Department of Marine Science and Applied Biology, University of Alicante, 03690 Alicante, Spain; forcada@ua.es (A.F.-A.); carlos.valle@ua.es (C.V.)
3   Centre d'Océanologie de Marseille, UMR CNRS 6540, Université de la Méditerranée, 13005 Marseille, France; mireille.harmelin@mio.osupytheas.fr
4   GIS Posidonie, Parc Scientifique et Technologique de Luminy, Case 901, 13288 Marseille, CEDEX 09, France; laurence.ledireach@univ-amu.fr
5   Parc Marin de la Côte Bleue, Observatoire, BP 42, 13620 Carry-le-Rouet, France; charbonnel.eric@parcmarincotebleue.fr
6   Departamento de Ecología e Hidrología, Campus de Espinardo, Universidad de Murcia, 30100 Murcia, Spain; jcharton@um.es (J.A.G.-C.); angelpr@um.es (A.P.-R.)
7   WWF-France, 6 Rue des Fabres, 13001 Marseille, France; dody@wwf.fr
8   Instituto Español de Oceanografía, Centro Oceanográfico de Baleares, Apdo. 291, 07080 Palma de Mallorca, Spain; olga.renones@ieo.csic.es
*   Correspondence: bayle@ua.es; Tel.: +34-965-90-3400 (ext. 2977)

**Abstract:** The aim of this paper is to provide evidence of the species-specific export of adult fishes for some species or group of species from six well-enforced Mediterranean marine protected areas (MPAs): Cerbère-Banyuls and Carry-le-Rouet in France and Medes, Cabrera, Tabarca, and Cabo de Palos in Spain. We estimated the distance at which spillover of those individual or groups of species occur by directly assessing the existence of gradients of biomass across the MPA boundaries by means of underwater visual census, asuming that such gradients will be specifics in terms of structure (sharpness, slope, and intercept) for every species and group of species. A significant "reserve effect" was observed for biomass of some of the individual and grouped species in all MPAs. Decreasing gradients of biomass differ among taxons and are not related with the insular nature of the location. Different gradients of biomass resulted from the interaction between species characteristics and some ecological and structural drivers, and we did not find regular patterns for each taxa among MPAs, even though the same species can exhibit different gradient structure and/or spillover distances in the same MPA depending on the orientation. Habitat patch distribution and continuity seems the most important environmental factor explaining the existence and pattern of gradients at species level, interacting with fish home range and fishing pressure close to the limits of the MPAs. Managers should take in account the surrounding distribution of habitats in terms of complexity and quality in order to optimize the spillover capacity of the MPAs.

**Keywords:** fish spillover; adult biomass; marine conservation; visual census; fisheries; management; impact; Mediterranean sea

## 1. Introduction

Over the last years, most mediterranean coastal fish species have been overexploited [1], raising doubts about the long-term sustainability of coastal fisheries [2]. In addition, fish habitats have been strongly altered by widely used impacting fishing gears (trawls, dredges, etc.), resulting in reduced seabed complexity and removal of macrobenthic organisms that

provide shelter for other species [3]. The poor performance of conventional fisheries management has led to increased interest on marine protected areas (MPAs) [4,5], because they are considered a potential solution to enhance the long-term sustainability of many fisheries. MPAs are widely promoted to restore fish species populations to benefit surrounding fisheries through two main mechanisms: net emigration of adult and juvenile fishes outside the MPA (spillover effect) and export of pelagic eggs and larvae from restored spawning stocks inside the MPA [6]. Previous studies have demonstrated positive effects of MPAs restoring some fish populations and on adjacent fisheries, by analyzing long-term data (e.g., [7]) or measuring gradients of biomass through the MPA border [8].

The gradient across marine reserve boundaries results of emigration, home range movements, and/or relocation of fish from MPAs [9]. Over an increasing distance from the MPA, the pattern of fish abundance and/or biomass should produce a gradient with greater negative slope as the diffusion process becomes more important and/or as the fishing pressure increases. The negative slope of the gradient can be interpreted as evidence of spillover of adult fish to surrounding fishing areas and its magnitude would be species-dependent. The existence of gradient due to spillover should be considered plausible given the reported direct evidences on the movements of some fishes from MPAs [10–14] or from other kind of regulated areas (e.g., fish farms; [15]), doing periodic wide excursions of up to 30 km to open areas, at least for species with a medium and large home range.

Mediterranean MPAs have been established mainly to protect zones that already harbor structurally complex habitats. Some studies conducted in Mediterranean MPAs demonstrate an increase in the abundance, biomass, and size of certain fish species due to protection [16–19]. Their role in sustaining local fisheries has been confirmed for some species (e.g., Norway lobster (*Nephrops norvegicus*) [20]. Furthermore, gradients of total fish abundance and biomass have been evidenced for several Mediterranean MPAs using a fishing-independent approach (see [21] for a review), remaining dependent on the analysis of fish biomass gradients at species or higher taxonomic level. Most studies on MPAs are performed on single locations; in this case, analyzing the effects of fish spillover on a time scale gives the best evidence on this process [9,22]; however, such studies need longer-term efforts compared to studies performed over horizontal spatial scales [23] and make it difficult to assess general trends in fish distribution across MPA boundaries [24,25]. For this reason, wider statements on fish spillover and inferences on the spatial extent of fish export require sampling fish assemblages in many sites across a gradient of distances from the reserve border, both outside and within the reserve, at several MPAs [23].

Although the data for this study were collected twenty years ago, the findings remain highly relevant due to the lack of similar studies examining directly biomass export from marine protected areas (MPAs) in the Mediterranean region. To date, no recent research has comprehensively addressed these processes in the Mediterranean, underscoring the enduring value of our results. These findings offer critical insights into biomass gradients and spillover patterns that continue to inform MPA management strategies, especially given the long-term ecological dynamics involved and the persistent need to optimize conservation practices within this unique marine ecosystem. The purpose of this paper is to provide evidence of the species-specific export of adult fishes for some species and group of species from six Mediterranean marine protected areas estimating the distance at which spillover of those individual or groups of species occur by directly assessing the existence of gradients of biomass across the MPA boundaries, asuming that such gradients will be specifics in terms of structure (sharpness, slope, and intercept) for every species or group of species.

## 2. Materials and Methods

### 2.1. Study Locations

The study was conducted on six Marine Protected Areas (MPAs) located in the Western Mediterranean. Two of these areas are in France (the Cerbère-Banyuls Marine Natural Reserve and the Carry-le-Rouet Marine Reserve in Côte Bleue Marine Park, referred to

hereafter as Banyuls and Carry-le-Rouet) and the other four are in Spain (Cabo de Palos-Islas Hormigas Marine Reserve, Cabrera Archipelago National Park, Medes Islands Marine Reserve, and Tabarca Island Marine Reserve, hereafter referred to as Cabo de Palos, Cabrera, Medes, and Tabarca). See Figure 1 for an overview of the study locations.

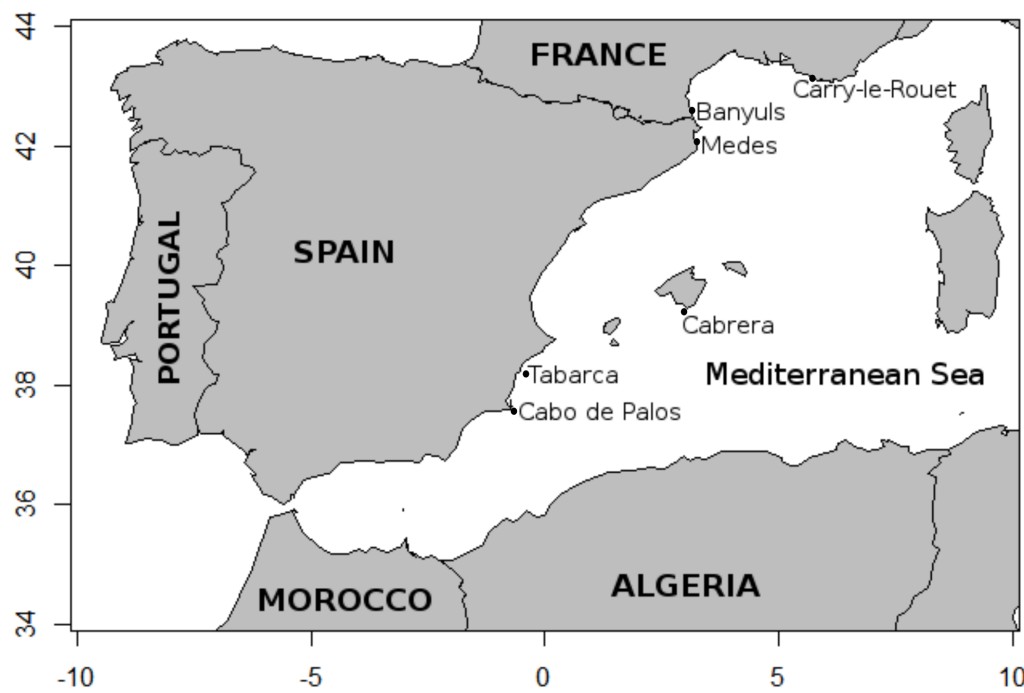

**Figure 1.** Localization of the six marine protected areas studied in France and Spain in the Mediterranean Sea.

The MPAs selected meet common criteria, such as being over 10 years old and maintaining a high level of protection, enforcement, and monitoring (Table 1). Some studies have shown that both the age of the reserve and strict regulatory compliance are key factors in evaluating the effectiveness of MPAs [26,27]. The long history of effective protection in these areas provides a suitable model for examining the effects of reserves on fish communities in the Western Mediterranean. The selected reserves are at least 10 years old, as studies have indicated that a steady state is not reached before 6 to 10 years after their establishment [28]. Among the six MPAs, three are located in mainland areas (Banyuls, Carry-le-Rouet, and Cabo de Palos) and the other three on islands (Cabrera, Medes, and Tabarca), allowing for an exploration of geographic isolation patterns. All MPAs include an integral reserve area (IR), where commercial and recreational fishing is prohibited, and, with the exception of Carry-le-Rouet, they also include a buffer zone (BZ), where some controlled fishing activities are allowed. In the case of the Tabarca Marine Protected Area, two distinct habitats were analyzed separately: rocky reef and Posidonia oceanica meadows. These habitats were treated as independent study zones due to their differing ecological characteristics, which made direct comparisons inappropriate. This approach allowed us to examine spillover effects in two contrasting habitats within the same MPA, enhancing the scope of the study.

**Table 1.** Main characteristics of the six MPAs studied in the northwestern Mediterranean, number and direction of fish gradients analyzed, maximum distance sampled outside the MPA, number of sectors, zones, replicates and total number of fish species recorded in each MPA. IR = integral reserve, BZ = buffer zone, Is = island, ML = main land, N = north, S = south, Max. distance = maximum distance sampled in fished areas from the MPA border.

| MPA | Country | Year of Creation | Location | Size (ha) | Level of Protection | Habitat Sampled | Nb Gradients | Direction of Gradients | Max. Distance | Nb of Sectors (In-Out MPA) | Nb Zones | Nb Replicates | Nb Fish Species |
|---|---|---|---|---|---|---|---|---|---|---|---|---|---|
| Banyuls | France | 1974 | ML | 650 | IR + BZ | Rocks | 2 | N + S | 5370 | 9 (3-6) | 27 | 162 | 43 |
| Cabo de Palos | Spain | 1995 | ML | 1898 | IR + BZ | Rocks | 2 | N + S | 8779 | 9 (3-6) | 27 | 162 | 48 |
| Cabrera | Spain | 1991 | Is | 8680 | IR | Rocks | 1 | N | 22,400 | 14 (5-9) | 42 | 252 | 51 |
| Carry-le-Rouet | France | 1983 | ML | 85 | IR | Rocks | 2 | E + W | 2668 | 9 (3-6) | 27 | 162 | 40 |
| Medes | Spain | 1983 | Is | 418 | IR + BZ | Rocks | 1 | N | 3058 | 10 (4-6) | 30 | 126 | 51 |
| Tabarca (rocks) | Spain | 1986 | Is | 1400 | IR + BZ | Rocks | 1 | N | 5448 | 7 (4-3) | 21 | 126 | 41 |
| Tabarca *Posidonia* | Spain | 1986 | Is | 1400 | IR + BZ | *Posidonia* beds | 1 | N | 1970 | 7 (4-3) | 21 | 126 | 39 |

## 2.2. Sampling Methodology

Fish species, along with their numbers and sizes, were documented on standardized recording sheets through visual underwater surveys conducted by trained scuba divers. Surveys took place over rocky substrates at depths of 6 to 12 m, using $25 \times 5$ m strip transects set parallel to the coastline. Due to the extensive *Posidonia oceanica* meadows surrounding Tabarca Island, this seagrass habitat was also included in the survey, with wider $50 \times 5$ m transects to account for the greater fish dispersal in this environment. All observed fish species were noted, with the exception of small, sedentary benthic species like blennies and gobies, and strictly pelagic species such as clupeids and engraulids, which are minimally affected by protective measures. Fish counts were documented up to 10 individuals, while larger groups were assigned to one of several abundance categories found in the scientific literature (11–30, 31–50, 51–200, 201–500, and >500 individuals), which is standard in visual census methods. The sizes of individual fish were recorded in 2-cm increments, and their weights were estimated based on established length–weight equations specific to Mediterranean fish.

Given that habitat characteristics significantly influence the spatial distribution of Mediterranean fish assemblages, main habitat features were systematically recorded on each transect. The investigation and measurement of patchiness were addressed measuring the habitat heterogeneity, which was visually assessed based on the percentage cover of substrates like rock, pebbles, sand, and *P. oceanica*, and habitat complexity, estimated by counting small, medium, and large rock formations, also assessing the maximum height of vertical structures and noting depth ranges.

Surveys were performed by the same team of trained scientific divers during the summer months, from June to October 2003 in Cabo Palos, Carry-le-Rouet, and Tabarca, and from June to September 2004 in Banyuls, Cabrera, and Medes. The warm season in the Mediterranean (June to October) provides optimal physical and environmental conditions for visual census, as fish populations exhibit their higher diversity, activity levels are at their peak in this period and their highest indicator value [29], providing a more comprehensive representation of the fish assemblages in the studied regions. While the sampling does not encompass other seasons, this approach was intentionally chosen to focus on the period when conditions are most favorable for capturing the diversity, dynamics, and ecological significance of the ichthyofaunal communities; conducting surveys within this timeframe minimizes seasonal variations, thus improving the reliability and comparability of observed spatial patterns in fish distribution.

## 2.3. Experimental Design

The sampling framework for each Marine Protected Area (MPA) was organized as follows: nine sectors, each spaced by thousands of meters, were established progressively further from the central zone of the MPA. These included three sectors within the MPA boundaries and six outside in fished regions, split equally in opposite directions. Within each sector, three zones were randomly selected at scales of several hundred meters, and in each zone, six transects were sampled, spaced apart by tens of meters (Figure 2). In MPAs situated on the mainland (Banyuls, Carry-le-Rouet, and Cabo de Palos), fish population gradients were examined in two opposite orientations. However, for MPAs around islands (Cabrera, Medes, and Tabarca), sampling was only feasible in the northward direction due to a lack of suitable rocky habitats on the southern sides of these islands. Over the 2003 and 2004 surveys, a total of 1026 underwater visual fish counts were conducted. This experimental design was carefully planned to minimize confounding effects caused by spatial variations and to ensure the comparability of results across the studied localities. This was achieved by employing a standardized protocol applied uniformly across all sampling sites, controlling for potential third-variable influences that could arise from differences in environmental or ecological conditions between locations. This approach was critical in ensuring that observed differences were attributable to genuine spatial variations in community structure rather than methodological inconsistencies or uncontrolled spatial

biases. Although the experimental design was developed and applied in 2003–2004, it remains—and will remain—fully valid and scientifically sound for addressing similar research questions in the future, allowing future comparisons over different sampling times.

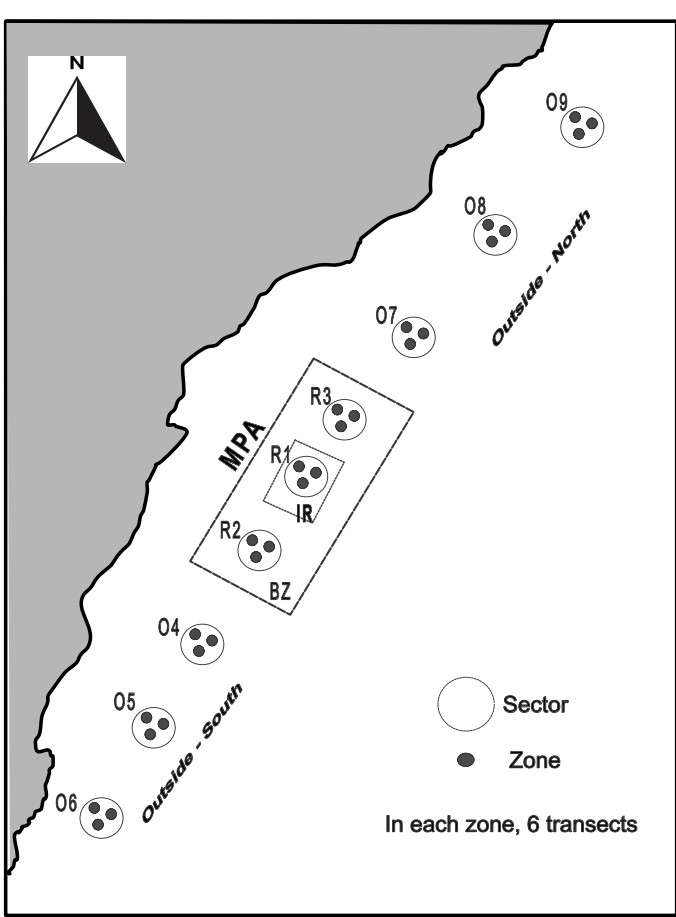

**Figure 2.** Spatially nested sampling design for studying fish biomass gradients across MPA boundaries. R = sectors located inside the marine reserve. O = sectors located outside the marine reserve. IR = integral reserve. BZ = buffer zone.

*2.4. Statistical Procedures for Assessing Spillover Effects*

The spillover effects of the selected MPAs was evaluated by means of statistical methods that ensure robust and reliable comparisons between inside and outside the MPAs and among the selected taxa. The analysis focused on quantifying spatial patterns of species biomass and their variation across the boundaries of the protected areas. To assess the effect of protection within Marine Protected Areas (MPAs), we evaluated average fish biomass across transects, adjusted for habitat type (125 m² for rocky substrates and 250 m² for *Posidonia oceanica* meadows) to observe distribution trends among taxa: big serranids (*Epinephelus caninus*, *E. costae*, *E. marginatus*, and *Mycteroperca rubra*), small serranids (*Serranus atricauda*, *S. cabrilla*, and *S. scriba*), big labrids (*Labrus bimaculatus*, *L. merula*, *L. viridis*, and *Symphodus tinca*), *Coris julis*, *Diplodus* spp. (*D. annularis*, *D. cervinus*, *D. puntazzo*, *D. sargus*, and *D. vulgaris*), other sparids (including *Dentex dentex*, *Sparus aurata*, and *Spondyliosoma cantharus*), *Mullus surmuletus*, and *Sciaena umbra*. To identify a clear "reserve effect", mean biomass values of taxons within MPAs and in adjacent fished areas were analyzed using variance analysis ($p < 0.05$) [30]. Variance homogeneity was tested via Cochran's test [31], and where necessary, data transformations ($\sqrt{(x)}$ or $ln(x + 1)$) were applied. When heterogeneity remained, analyses proceeded on untransformed data, as balanced experimental designs with large sample sizes generally ensure the robustness of variance analysis [30].

Since Mediterranean MPAs are often located in complex natural habitats that promote diverse fish communities, it was essential to distinguish the influence of habitat structure from that of protection itself. To this end, we first isolated the variance attributable to habitat descriptors using multiple regression models (GLMs [32]). The relationship between habitat complexity and fish biomass was, thus, explored using predictors recorded for each transect. Outliers were identified and excluded prior to analysis. To further evaluate spatial patterns, we examined biomass trends in relation to distance from MPA boundaries using generalized additive models (GAMs). The zero point represented the edge of the integral reserve (IR), with zones inside the IR having negative distances and external zones having positive distances. GAMs were employed to model these residuals in relation to distance, ensuring that any observed gradient was a result of protection rather than habitat variability. A loess smoother was applied to model the distance to IR boundary as a continuous variable. The distance at which "spillover" effects might become evident was determined by identifying the minimum spatial separation between zones where a marked difference in mean biomass appeared across the IR boundary. This threshold was determined for each species or group showing significant negative correlations across the gradient.

These statistical procedures were designed to capture differences in fish biomass between insed and outside the MPAs, and spatial gradients and differences attributable to spillover effects, while accounting for potential confounding factors due to spatial variability at different scales. By applying these methods, the study ensures that the observed patterns are both statistically and ecologically meaningful. The results will provide a valuable baseline for understanding the effectiveness of MPAs and their capacity to influence adjacent areas.

## 3. Results

### 3.1. "Reserve Effect" Analysis on Fish Biomass Across MPAs: Comparison of "In" and "Out" Data

To assess the potential "reserve effect" of the marine protected areas (MPAs), the study compared ichthyofaunal asemblages observed in sectors within the MPAs ("In" data) and in sectors adjacent, unprotected areas ("Out" data). The experimental design specifically accounted for spatial comparability between sampling locations. Sectors classified as "In" were located within the clearly defined boundaries of the MPAs, while "Out" sectors were positioned immediately outside these boundaries in areas with similar habitat types, depth ranges, and environmental conditions to minimize confounding variables. The sampling methodology described in Section 2.2 ensured uniformity in data collection across all locations; visual censuses were conducted along a balanced number of transects across sampling levels, both inside and outside the MPAs. The design explicitly aimed to evaluate spatial differences attributable to protection status while controlling for ecological variability unrelated to management. This approach was fundamental for isolating the effects of MPA boundaries and detecting gradients in species richness, abundance, and biomass indicative of "spillover" effects.

A notable "reserve effect" was detected for the biomass of several taxa across all MPAs (Table 2). In Banyuls, Cabo de Palos, Medes, and Tabarca Rocks, large serranids showed significantly higher biomass values within MPA boundaries. Biomasses of small serranids and *Diplodus* spp. were elevated in all MPAs, although the differences were less pronounced compared to those of large serranids. Large labrids exhibited significantly greater biomasses in five MPAs. Other sparid species displayed significantly higher biomasses exclusively within the Banyuls, Cabo de Palos, and Carry MPAs, with less pronounced differences between protected and unprotected sectors than *Diplodus* spp. due to their more solitary habits and lower occurring densities. Biomass of *Coris julis* was significantly higher in Cabo de Palos, Medes, and Tabarca-Posidonia, whereas *Mullus surmuletus* showed increased biomass across five MPAs, reaching significance only in Carry. In Tabarca *Posidonia*, all species or groups analyzed—except *Sciaena umbra*—demonstrated higher biomasses within the protected zones.

**Table 2.** Mean (±SD) biomass in grams for individual and grouped species per transect recorded inside (In) and outside (Out) the six MPAs studied with results of ANOVAs.

| MPA | | Big Serranids | Small Serranids | Big Labrids | *C. julis* | *Diplodus* spp. | Other Sparids | *M. surmuletus* | *S. umbra* |
|---|---|---|---|---|---|---|---|---|---|
| Banyuls | In | 2488 ± 1312 | 97 ± 10 | 298 ± 42 | 144 ± 14 | 2726 ± 442 | 9473 ± 3769 | 92 ± 18 | 189 ± 94 |
| | Out | 0 | 80 ± 7 | 222 ± 24 | 155 ± 12 | 1073 ± 118 | 2140 ± 679 | 115 ± 20 | 0 |
| | F-test | 7.237 ** | 1.659 ns | 2.731 ns | 0.313 ns | 21.766 *** | 6.732** | 0.52 ns | 8.102** |
| Cabo Palos | In | 12,950 ± 2440 | 71 ± 11 | 60 ± 23 | 117 ± 18 | 2431 ± 343 | 1134 ± 357 | 5 ± 3 | 850 ± 232 |
| | Out | 151 ± 103 | 42 ± 4 | 141 ± 16 | 75 ± 6 | 478 ± 89 | 77 ± 24 | 19 ± 5 | 7 ± 3 |
| | F-test | 54.949 *** | 8.781 ** | 7.856 ** | 7.209 ** | 50.948 *** | 17.271 *** | 2.818 ns | 26.365 *** |
| Cabrera | In | 2751 ± 493 | 122 ± 10 | 232 ± 33 | 92 ± 9 | 2699 ± 687 | 1894 ± 1887 | 34 ± 13 | 371 ± 144 |
| | Out | 3764 ± 787 | 106 ± 6 | 144 ± 13 | 101 ± 7 | 1809 ± 254 | 58 ± 41 | 26 ± 4 | 61 ± 15 |
| | F-test | 0.75 ns | 2.078 ns | 8.34 ** | 0.427 ns | 2.175 ns | 1.901 ns | 0.511 ns | 8.892 ** |
| Carry | In | N.P. | 84 ± 12 | 405 ± 70 | 226 ± 23 | 1109 ± 299 | 16 ± 10 | 23 ± 9 | 79 ± 54 |
| | Out | N.P. | 42 ± 6 | 154 ± 24 | 146 ± 17 | 264 ± 39 | 3 ± 1 | 14 ± 5 | 0 |
| | F-test | | 12.94 *** | 16.88 *** | 0.194 ns | 15.044*** | 6.243 * | 6.754 ** | 4.352 * |
| Medes | In | 5956 ± 1129 | 101 ± 9 | 410 ± 55 | 112 ± 10 | 3636 ± 666 | 679 ± 345 | 40 ± 14 | 542 ± 220 |
| | Out | 0 | 50 ± 5 | 71 ± 18 | 62 ± 7 | 1038 ± 184 | 8 ± 3 | 10 ± 3 | 0 |
| | F-test | 20.8 *** | 17.671 *** | 26.838 *** | 12.525 ** | 10.906 ** | 2.817 ns | 3.249 ns | 4.514 * |
| Tabarca rocks | In | 1677 ± 413 | 112 ± 10 | 251 ± 33 | 72 ± 7 | 1935 ± 450 | 1895 ± 450 | 7 ± 2 | 295 ± 91 |
| | Out | 177 ± 97 | 99 ± 13 | 154 ± 30 | 60 ± 5 | 650 ± 74 | 2408 ± 657 | 4 ± 1 | 74 ± 39 |
| | F-test | 9.55 ** | 0.603 ns | 4.232 * | 1.412 ns | 8.956 ** | 0.443 ns | 0.537 ns | 3.954 * |
| Tabarca Posid. | In | 861 ± 611 | 98 ± 12 | 186 ± 43 | 103 ± 10 | 466 ± 109 | 492 ± 122 | 17 ± 7 | 3 ± 2 |
| | Out | 0 | 41 ± 4 | 26 ± 7 | 28 ± 5 | 102 ± 15 | 228 ± 70 | 1 ± 1 | 4 ± 4 |
| | F-test | 1.488 ns | 13.761 *** | 10.429 ** | 30.749 *** | 8.135 ** | 2.954 ns | 3.041 ns | 0.004 ns |

N.P.: not present; ns: not significant; (*): $p < 0.05$; (**): $p < 0.01$; (***): $p < 0.001$.

### 3.2. Accounting for Habitat Patchiness Influence in MPA Fish Distribution

To clarify the role of protection in fish biomass distribution, our analysis accounted for habitat effects, ensuring that observed patterns could be attributed to protection status rather than habitat variability. This approach allowed us to isolate the influence of marine protected areas on biomass gradients, while still recognizing that some species are inherently more sensitive to small-scale habitat differences than others. Results exhibited that, despite the overall visual uniformity of the environment, subtle variations in habitat structure were observed, which played a critical role in shaping the ichthyofaunal assemblages. These structural differences, such as variations in substrate composition, the presence of seagrass patches, rocky outcrops, or small depressions, provided microhabitats that supported different species and ecological functions. Multiple linear regression analyses on the biomass of selected taxa (Table 3) indicated that habitat structure significantly influenced the distribution of all species groups in Cabo de Palos, while having a weaker effect in Banyuls, Cabrera, and Carry. Results from the *Posidonia* beds at Tabarca were particularly notable; in what seems like a homogeneous area, it significantly affected biomass distribution for most fish groups.

Among the taxa studied, *Coris julis*, *Diplodus* spp., and large labrids exhibited significant responses to habitat variations, even within the relatively uniform seabed of *Posidonia oceanica*. Subtle habitat features, such as patchiness in seagrass density, proximity to rocky edges, or slight depth variations, may explain the differential responses of these species. In contrast, *Mullus surmuletus* and other sparid species appeared less influenced by these habitat characteristics, displaying more consistent biomass distributions across sites.

**Table 3.** Summary of multiple linear regressions (GLMs) of mean biomass of individual and groups of species vs. all habitat characteristics together. Values of adjusted $R^2$ and the level of significance of each analysis were indicated.

| MPA | Big Serranids | Small Serranids | Big Labrids | *Coris julis* | *Diplodus* spp. | Other Sparids | *M. surmuletus* | *S. umbra* |
|---|---|---|---|---|---|---|---|---|
| Banyuls | ns | ns | 0.201 *** | 0.266 *** | 0.350 *** | ns | ns | ns |
| Cabo Palos | 0.545 *** | 0.109 * | 0.322 *** | 0.101 * | 0.562 *** | 0.285 *** | 0.176 ** | 0.369 *** |
| Cabrera | 0.247 * | ns | ns | 0.279 ** | ns | 0.254 * | ns | ns |
| Carry | N.P. | ns | 0.316 ** | 0.243 * | 0.246 * | ns | ns | ns |
| Medes | ns | ns | 0.243 *** | 0.297 *** | 0.270 *** | ns | 0.119 *** | ns |
| Tabarca rocks | 0.243 ** | 0.131 * | ns | 0.198 ** | 0.142 * | ns | ns | ns |
| Tabarca Posidonia | 0.218 * | 0.185 ** | 0.265 *** | 0.202 ** | 0.344 *** | ns | 0.329 *** | 0.185 ** |

N.P.: not present; ns: not significant; (*): $p < 0.05$; (**): $p < 0.01$; (***): $p < 0.001$.

### 3.3. Species-Specific Spillover Patterns

Correlation coefficients of biomass for the selected taxa with distance from the MPA, along with their levels of significance, are presented in Table 4. Negative values indicate a decrease in fish biomass from the MPA towards fished areas, whereas positive values suggest the opposite trend. Except for *M. surmuletus*, most fish groups analyzed showed a generally negative correlation between their biomass and distance from the MPA core. These negative correlations were statistically significant for large and small serranids, *S. umbra*, and *C. julis*. The more mobile *Diplodus* species also responded strongly to protection, whereas other sparids were less sensitive. *Mullus surmuletus* tended to show positive correlation values more frequently, but this pattern was significant only in Banyuls. Within MPAs, the same species exhibited different biomass correlation values depending on direction (e.g., large serranids and large labrids in Cabo de Palos, or small serranids, large labrids, and *C. julis* in Carry).

Results of GAMs on the biomass residuals for the taxa analyzed, as a function of distance to reserve boundaries, revealed significant non-linear relationships in 18 out of 80 cases analyzed (Table 5). The variance explained by these models was generally low, with a maximum of 28.2% in Cabrera for large serranids. Fitted values for the GAMs showed four main patterns of biomass gradients:

(i)      Mean biomass declined abruptly at the IR-BZ boundary (Figure 3a,c,d,g,l,m,o–r);

(ii)      Mean biomass declined between IR and BZ, increasing outside the MPA near the external boundaries (Figure 3b,j,k,n);

(iii)      Biomass increased between the IR and BZ boundaries, declined immediately outside the MPA, and then increased farther into the fished area (Figure 3f,h);

(iv)      Biomass increased outside the MPA (Figure 3e,h).

**Table 4.** Values for linear correlation of fish biomass calculated on residuals after extracting the effects of habitat related with distance from IR.

| MPA | Orientation | Big Serranids | Small Serranids | Big Labrids | *Coris julis* | *Diplodus* spp. | Other Sparids | *M. surmuletus* | *S. umbra* |
|---|---|---|---|---|---|---|---|---|---|
| Banyuls | North | −0.276 ** | −0.141 ns | 0.198 * | 0.049 ns | −0.352 *** | 0.014 ns | 0.092 ns | −0.181 ns |
| Banyuls | South | −0.327 *** | −0.160 ns | 0.078 ns | 0.027 ns | −0.135 ns | 0.027 ns | 0.279 ** | −0.245 * |
| Cabo Palos | North | −0.174 ns | −0.048 ns | 0.345 *** | −0.054 ns | −0.054 ns | 0.027 ns | −0.030 ns | −0.186 ns |
| Cabo Palos | South | 0.043 ns | 0.034 ns | 0.150 ns | −0.011 ns | 0.046 ns | −0.001 ns | 0.120 ns | −0.104 ns |
| Cabrera | | −0.017 ns | 0.021 ns | 0.076 ns | 0.062 ns | 0.011 ns | 0.021 ns | −0.102 ns | 0.043 ns |
| Carry | East | N.P. | −0.432 *** | −0.355 *** | −0.321 ** | 0.008 ns | −0.068 ns | −0.018 ns | −0.169 ns |
| Carry | West | N.P. | 0.024 ns | −0.105 ns | 0.058 ns | −0.220 * | 0.025 ns | −0.079 ns | −0.118 ns |
| Medes | | −0.384 *** | −0.144 ns | −0.278 *** | −0.217 * | −0.284 ** | −0.157 ns | 0.096 ns | −0.182 * |
| Tabarca rocks | | −0.154 ns | 0.288 ** | 0.021 ns | −0.169 ns | −0.041 ns | −0.041 ns | 0.033 ns | −0.041 ns |
| Tabarca Posidonia | | −0.001 ns | −0.025 ns | −0.214 * | −0.332 *** | −0.097 ns | −0.198 * | −0.134 ns | 0.005 ns |

The correlation coeficient is given with its statistical significance. N.P.: species or group not present. *, $p < 0.05$; **, $p < 0.01$; ***, $p < 0.001$; ns: not significant.

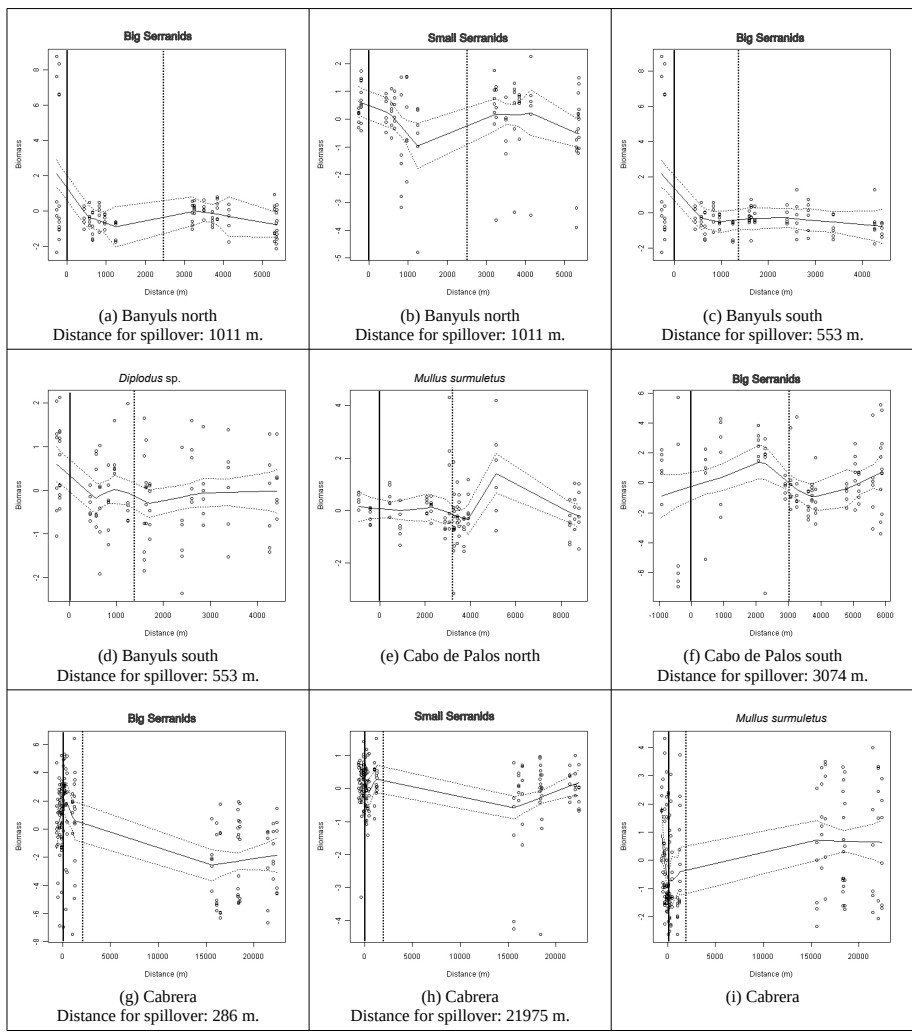

**Figure 3.** *Cont.*

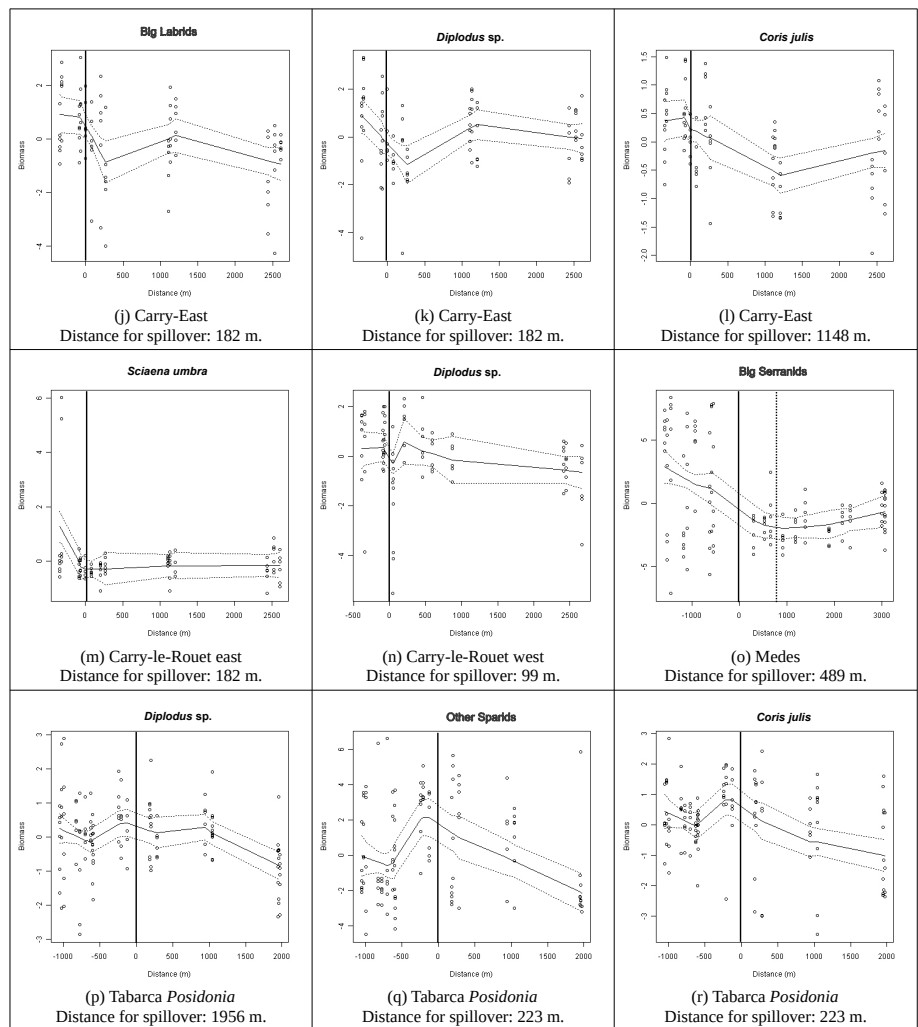

**Figure 3.** Fitted mean residuals of biomass for the considered taxa as a function of the smooth variable distance (m) from the integral reserve boundary derived from significant generalized additive models (GAMs), in the six Mediterranean MPAs studied. The y-axis is scaled so that zero corresponds to the mean in the log scale. Dashed lines indicate mean ± 2 standard errors. The vertical line indicates the limit of the integral reserve (IR) and the vertical dashed line the limits of the buffer zone (BZ).

**Table 5.** Variance explained and significance values of the generalized additive models (GAMs) fitted with the residuals of fish biomass. Between brackets, mean distance for spillover for the considered taxa with negative gradient.

| MPA | Orientation | Big Serranids | Small Serranids | Big Labrids | *Diplodus* spp. | Other Sparids | *Coris julis* | *M. surmuletus* | *S. umbra* |
|---|---|---|---|---|---|---|---|---|---|
| Banyuls | North | 0.243 *** (1011 m) | 0.110 * (1011 m) | 0.07 | 0.18 | 0.05 | 0.06 | 0.07 | 0.1 |
| Banyuls | South | 0.246 *** (553 m) | 0.09 | 0.07 | 0.097 * (553 m) | 0.05 | 0.06 | 0.1 | 0.12 |
| Cabo Palos | North | 0.105 | 0.022 | 0.186 | 0.038 | 0.053 | 0.078 | 0.143 ** | 0.084 |
| Cabo Palos | South | 0.094 * (3074 m) | 0.04 | 0.06 | 0.02 | 0.03 | 0.07 | 0.07 | 0.06 |
| Cabrera |  | 0.282 * (286 m) | 0.082 * (21975 m) | 0.05 | 0.13 | 0.03 | 0.08 | 0.129 * | 0.02 |
| Carry | East | N.P. | 0.24 | 0.220 * (99 m) | 0.154 ** (182 m) | 0.07 | 0.230 ** (182 m) | 0.04 | 0.209 ** (182 m) |

**Table 5.** *Cont.*

| MPA | Orientation | Big Serranids | Small Serranids | Big Labrids | *Diplodus* spp. | Other Sparids | *Coris julis* | *M. surmuletus* | *S. umbra* |
|---|---|---|---|---|---|---|---|---|---|
| Carry | West | N.P. | 0.09 | 0.06 | 0.175 * (99 m) | 0.1 | 0.06 | 0.06 | 0.11 |
| Medes | | 0.260 *** (489 m) | 0.04 | 0.09 | 0.12 | 0.07 | 0.06 | 0.04 | 0.07 |
| Tabarca rocks | | 0.030 | 0.145 | 0.017 | 0.032 | 0.059 | 0.060 | 0.016 | 0.020 |
| Tabarca *Posidonia* | | 0 | 0.02 | 0.1 | 0.168 ** (223 m) | 0.224 *** (223 m) | 0.191 * (223 m) | 0.04 | 0.04 |

N.P.: species or group not present. *, $p < 0.05$; **, $p < 0.01$; ***, $p < 0.001$; ns: not significant.

### 3.4. Distance Metrics for Fish Spillover

The estimated global mean distance for fish spillover from the IR boundary was 2271 ± 595.18 m, encompassing the sixteen significant negative relationships identified in the GAMs. These distances ranged from a minimum of 99 m for *Diplodus* sp. (Figure 3n) at Carry West to a maximum of 21,975 m for small serranids at Cabrera (Figure 3h). The GAMs visually confirmed these estimates, demonstrating a pronounced depletion in biomass near the IR boundaries for most of the taxa considered. Consequently, the mean spillover distance estimated in this study likely occurs on a scale of tens to hundreds of meters in the majority of the six MPAs analyzed.

## 4. Discussion

### 4.1. Impact of MPAs on Biomass Enhancement

Higher mean biomass values were observed within the six MPAs for most taxa considered, with substantial and statistically significant differences in 57% of cases analyzed. This outcome reflects a combination of increased abundances and larger individual sizes of these species within the MPAs, as previously documented in each region by other studies [16–19]. These findings support the evidence that Mediterranean MPAs contribute to biomass increases in certain fish species, a trend also observed in many tropical and temperate MPAs (e.g., [25,33]). The protective effects are generally more pronounced in top predators, as shown in previous studies (e.g., [34,35]), and/or in species significantly impacted by fishing, highlighting the lasting impacts of protection.

### 4.2. Drivers of Species-Specific Gradients of Biomass

Results revealed that biomass gradients were generally non-linear, with sharp declines near MPA boundaries in most cases. The four primary biomass change patterns observed across distances in this study cannot be exclusively attributed to specific species. Instead, these patterns relate to various structural characteristics, individually or in combination, specific to each MPA. High percentages of negative correlations were observed across different species groups, indicating a general biomass decrease from MPAs to adjacent fished areas. However, only 23% of the linear correlations between groups and distance showed significantly negative trends, a smaller percentage compared to results for overall assemblage descriptors [21]. Biomass gradient declines appear to depend on MPA-specific factors for each species group, given that results were not homogeneous across MPAs or taxa. Non-significant patterns observed for certain highly targeted groups (e.g., large serranids, *Diplodus* spp., and other sparids) in some locations may be due to differences in habitat characteristics between designated 'no-take' zones and regulated or open areas. This is despite overall significantly higher biomass levels within MPAs. While this study's fine-scale spatial data provide high comparability, multiple linear regressions confirmed significant relationships between habitat characteristics and fish biomass distribution, underscoring the need to account for habitat variability when assessing gradients related to protection [36].

Notably, the inverse *"reserve effect"*—with biomass positively correlated with distance—was observed in cases such as large and small serranids in Cabrera, *Diplodus* spp., other sparids,

and *C. julis* in Tabarca, where these taxa were consistently less abundant within IR zones and increased in biomass with distance from IR boundaries. One plausible explanation for this pattern is habitat patchiness influence within and around the MPAs, as biomass increases closer to boundaries often align with heterogeneous rocky patches (e.g., large serranids in Cabrera). Habitat complexity and patch spatial distribution, especially discontinuities, likely influence both assemblage structure and fish distribution [37], thereby affecting the strength (or even existence) of biomass spatial gradients. Evidence from Tabarca supports this, where different gradients emerged for the same species depending on habitat characteristics (relative abundance of *Posidonia* beds and rocky substrates). Additionally, processes such as competition or predation, particularly for piscivorous species heavily targeted by fishing, may deplete prey species populations, modifying gradient patterns for these species [38]. Alternative explanations for positive gradient patterns may include increased fishing effort in RU zones and unprotected areas near MPA boundaries [39], observed in the six MPAs studied, and factors such as certain taxa's moderate-to-high mobility [40], greater post-settler spillover, larval dispersal over long distances, density-dependent displacement by more competitive species within MPAs, and habitat preference [13]. Additionally, 'trophic stress' may be reduced within MPAs if primary and secondary productivity are enhanced [41].

Intense fishing pressure around MPAs can drive spillover [42]. Fishing effort surrounding the six MPAs was comparable and mainly associated with MPA proximity and 'fishing-the-line' practices. This often results in abrupt boundary biomass declines [9] that correlate with fishing effort data. While fishing effort concentration along MPA boundaries is sometimes considered evidence of spillover [23], much of this spatial distribution actually aligns with habitat patch distribution and its associated fish assemblages. This reinforces the critical role of habitat structure in shaping gradient patterns. Other ecological processes likely act uniformly across MPAs, making it improbable that these processes selectively influence specific MPAs to produce the observed differences.

Fish mobility likely impacts the existence and structure of biomass gradients [9,40], with gradient profiles varying based on species behavior. Gradients tend to be sharper for low-mobility species or those with high catchability. In certain cases (e.g., Banyuls north and south, Medes), large serranids exhibited sharp biomass declines; however, the pattern was reversed in others (e.g., Cabo de Palos south, Cabrera). Explaining these differences is challenging, as fish mobility and fishing effort appear consistent across the MPAs. While differing movement rates among smaller groupers may contribute [13], habitat complexity and seascape features within and around MPAs seem the most plausible explanation. For *M. surmuletus*, sharper gradients and significantly higher biomasses within MPAs were anticipated due to the limited mobility observed in other family members [43]. However, this was not the case, suggesting that habitat choice in this study and the typical habitat of *M. surmuletus* (sandy grounds) likely contributed to biomass increases outside MPAs, where this habitat was more prevalent.

*4.3. Influence of Species-Specific Characteristics on Spillover Distances*

Similar spillover distances would be expected for the same species across different MPAs, and consistently across all directions within a single MPA. Species with wider home ranges, such as *Diplodus* spp. and other sparids, should theoretically exhibit greater spillover distances than species with more sedentary bottom-dwelling habits (e.g., large and small serranids, large labrids, and *S. umbra*). This pattern should appear consistently within and among MPAs. However, our results diverged from this hypothesis: spillover distances varied between MPAs for the same species and even between different directions within the same MPA. In some cases, the same spillover distances were observed across taxa despite their different home range characteristics (e.g., large serranids vs. *Diplodus* spp. in Banyuls south, or *Diplodus* spp., *C. julis*, and *S. umbra* at Carry-le-Rouet), suggesting that ecological processes other than home range characteristics may play a more substantial role. Habitat heterogeneity is likely the primary factor driving these site-specific outcomes.

The predominant scale of spillover observed was frequently small, with most significant biomass gradients declining sharply within 500 m of the IR. Similar findings have been reported for other temperate [44,45] and tropical MPAs [22,46,47], where spillover distances typically occur within a few hundred meters, regardless of MPA size and age. However, the positive effects of spillover on fisheries beyond MPA boundaries or regulated areas (e.g., fish farms) have been directly observed in only a few cases, such as for certain carangids [48]. Movements are hypothesized to occur among juveniles and subadults due to density-dependent mechanisms, whereby the presence of older, more territorial individuals within protected areas displaces younger fish to surrounding areas [25], resulting in small-scale relocation (hundreds of meters), as reflected in our findings. Evidence of increased movement rates in smaller groupers [13] supports this hypothesis and may be influenced by the availability of suitable juvenile habitat throughout their ontogeny [49] and by habitat continuity [37,50]. Globally, most MPAs are designed to encompass predominantly high-complexity rocky habitats, often excluding less structured sandy areas that form natural barriers. This likely contributes to the small-scale movements (10–100 s of meters) documented around MPA boundaries for both bottom-dwelling species [14,51,52] and highly mobile pelagic species [11,12,53–55] across different regions, despite some species being capable of covering long distances (1000–10,000 s of meters; [15,56,57]). Home range also varies considerably in size and location throughout the year [58], and the timing of movement sampling can substantially affect observed patterns. Thus, both habitat distribution and methodological constraints may interact, potentially obscuring the true effects of protection on spillover dynamics.

## 5. Conclusions

In this study, we analyzed data from six Mediterranean MPAs, representing a range of ecological and management contexts. While the sample size is limited, the spatial scale and variability of these areas provide a robust framework for understanding general patterns of protection and spillover effects, avoiding pseudreplication among spatial scales and acchieving the analysis of the "reserve effect" over other colateral variables such as habitat characteristics. Similar patterns have been documented in recent studies [59,60], supporting the broader applicability of our findings. However, we acknowledge that additional studies involving a larger number of MPAs across more diverse conditions are needed to further validate and generalize these conclusions. Expanding the geographical and ecological scope of future research will be essential to refining our understanding of the effectiveness of MPAs in promoting biodiversity and fish biomass recovery.

Despite being based on data collected two decades ago, this study remains a unique and valuable source of empirical insights into species-specific biomass export from MPAs in the Mediterranean, a topic lacking in recent research. Our findings contribute to understanding biomass gradients and spillover dynamics, offering essential guidance for MPA management and conservation in the region. The study, covering six MPAs over a broad geographical range, represents the largest empirical direct effort to date for quantifying biomass gradients across species groups and serves as a generalizable framework for the Mediterranean region. Evidence from our findings indicates that biomass gradients vary among species groups and are influenced by the interaction of species traits with habitat patchiness distribution, rather than by the insular or continental nature of MPA locations. While habitat patchiness and continuity emerge as primary environmental factors shaping species-specific biomass gradients, these factors interact with fish mobility and fishing pressure to produce unique patterns in each MPA.

Practical recommendations for MPA management include considering habitat configuration, complexity, and quality to enhance biomass spillover. Strategic planning of fishing effort in adjacent zones based on observed biomass gradients could further support sustainable resource use. Given the findings' potential applicability, each MPA requires locally adapted management strategies that account for specific habitat structures and species compositions. Our results underscore the need for recent studies examining biomass spillover

under changing environmental conditions and human pressures, which could validate and expand on these baseline findings. Such studies would provide insights into how climate change and anthropogenic factors are reshaping biomass export, laying a foundation for adaptive MPA management across both Mediterranean and temperate marine zones.

**Author Contributions:** Conceptualization, J.T.B.-S., J.A.G.-C., Á.P.-R. and M.L.H.-V.; methodology, J.T.B.-S., A.F.-A., P.S.-J., L.L.D., E.C., J.A.G.-C., D.O., O.R., Á.P.-R. and M.L.H.-V.; formal analysis, J.T.B.-S., A.F.-A., J.A.G.-C. and M.L.H.-V., investigation, J.T.B.-S., A.F.-A., P.S.-J., L.L.D., E.C., J.A.G.-C., D.O., C.V., Á.P.-R. and M.L.H.-V.; resources, J.T.B.-S., L.L.D., E.C., J.A.G.-C., O.R. and Á.P.-R.; data curation, J.T.B.-S., A.F.-A., J.A.G.-C. and M.L.H.-V.; writing—original draft preparation, J.T.B.-S.; writing—review and editing, J.T.B.-S. and A.F.-A.; visualization, J.T.B.-S. and A.F.-A.; supervision, J.T.B.-S.; funding acquisition, J.T.B.-S., P.S.-J., L.L.D., E.C., J.A.G.-C., D.O., O.R., Á.P.-R. and M.L.H.-V. All authors have read and agreed to the published version of the manuscript.

**Funding:** This work was developed in the framework of the UE project BIOMEX (QLRT-2001-0891), WP2.

**Institutional Review Board Statement:** Ethical review and approval were waived for this study, due to the use of no sampling-harm techniques.

**Informed Consent Statement:** Not applicable.

**Data Availability Statement:** Data are contained within the article.

**Acknowledgments:** We thank the support of the coordinator Serge Planes, the BIOMEX group, and the staff of the six MPAs investigated. We are gratefull to Mikel Zabala for assistance in field work at the Medes Islands Marine Reserve.

**Conflicts of Interest:** The authors declare no conflicts of interest. The supporters had no role in the design of the study; in the collection, analyses, or interpretations of data; in the writing of the manuscript or in the decision to publish the results.

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
