# Peer review of "Species-Specific Spillover Patterns Detected by Biomass Gradients in Mediterranean Marine Protected Areas"

_sustainability, doi:10.3390/su162411089_

Round 1
Reviewer 1 Report
Comments and Suggestions for Authors
detailed comments in attachment

Author Response
General comment: This paper, based on field survey results from Mediterranean Marine Protected Areas (MPAs), analyzes the protective effects of MPA establishment and management on specific species or species groups. Overall, the paper is of low quality and far below the standard required for publication in an academic journal. Rejection is recommended.
Answer:
Thank you for your general feedback regarding our manuscript. While we respect your opinion and appreciate your review, we would like to respectfully highlight the contributions and significance of this study, as well as address your concerns.
Our research provides a comprehensive analysis of the protective effects of Mediterranean Marine Protected Areas (MPAs) on specific species and species groups. The study is based on field data collected using standardized methodologies and aims to address key ecological and management questions that are highly relevant to the ongoing global discussion on marine conservation.
We believe this work contributes meaningful insights by:
-
Providing empirical evidence on the ecological benefits of MPAs, specifically focusing on spillover effects and habitat-mediated responses of fish communities, which are critical for informing effective management practices.
-
Using a robust analytical framework that controls for habitat variability, ensuring that the observed patterns are attributable to protection status rather than confounding factors.
-
Offering a valuable baseline for future studies, given the temporal gap in monitoring efforts in the regions studied.
We acknowledge that the data were collected some time ago, which may contribute to your perception of the study’s relevance. However, the historical nature of the dataset adds value by allowing us to evaluate MPA impacts during an earlier stage of implementation, which can be used for comparison with more recent data.
We respectfully request specific feedback on the aspects you consider of "low quality" or below academic standards, as this would help us address your concerns more directly and improve the manuscript accordingly. We are committed to ensuring the scientific rigor and clarity of our work and are open to incorporating constructive suggestions.
Thank you again for your time and effort in reviewing our manuscript. We look forward to receiving your detailed feedback.
Specific comment 1:
The field survey data are from 20 years ago. What progress has been made in this research field over the last two decades? The paper lacks a comprehensive literature review, making it difficult to evaluate its relevance in this research direction.
Answer to specific comment 1:
Thank you for your comment regarding the age of the data and the perceived lack of a comprehensive literature review. We appreciate the opportunity to clarify these aspects and to emphasize the relevance of our study. Indeed, as mentioned in the manuscript, the data were collected 20 years ago. However, to date, there are no comparable datasets in the Mediterranean that provide specific empirical data on species-level spillover effects at this spatial scale. While progress has been made in analyzing spillover processes in other seas and for other species (e.g., https://doi.org/10.1038/s41598-021-82371-5), similar advancements have not been achieved in the Mediterranean. For this region, we identified a recent study (https://doi.org/10.1007/s41208-022-00396-7) that discusses factors influencing spillover but does not provide novel empirical data beyond what was already addressed in reference 32 of our manuscript. Regarding the comprehensive review of the field, we believe this is effectively addressed in the manuscript by citing the work of Di Lorenzo et al. (2016; reference 31), which provides a thorough bibliographic review of the Mediterranean context up to that point. This work summarizes the research landscape without adding specific empirical data, which reinforces the importance of our study in filling this gap. Including an additional extensive literature review would, in our view, unnecessarily expand the manuscript without adding significant value. The relevance of our manuscript lies in providing empirical data on species-specific spillover effects across a considerable spatial scale, using a replicable sampling protocol designed to isolate the effects of protection from confounding variables. This approach and the resulting findings adhere to the fundamental paradigms of empirical science, offering a solid contribution to understanding the role of marine protected areas in the Mediterranean. We hope this explanation addresses your concerns, and we remain open to further suggestions to enhance the manuscript.
Specific comment 2:
The analysis is based on data from only six experimental areas. Are the conclusions generalizable? This requires further discussion in the paper, with supporting evidence from recent publications. Without this, the conclusions lack persuasiveness.
Answer to specific comment 2:
Thank you for your comment regarding the generalizability of our conclusions based on data from six experimental areas. We appreciate the opportunity to address this concern and provide additional context.
Our study focuses on six marine protected areas (MPAs) selected to represent a range of environmental conditions, management effectiveness, and habitat types typical of Mediterranean MPAs. While the sample size is limited to these six areas, the spatial scale and variability encompassed by these sites offer valuable insights into general patterns of protection and spillover effects. Furthermore, the use of a standardized, replicable sampling protocol ensures that the results are robust and comparable, providing a foundation for broader ecological inferences.
To address your concern about generalizability, we agree that further discussion is warranted. We have added the following points to the manuscript:
-
Relevance of the study areas: The six MPAs included in this study cover a range of ecological and management contexts, increasing the applicability of the findings to other Mediterranean MPAs with similar characteristics. This is supported by the more recent review by Di Lorenzo et al., 2016.
-
Comparison with other research: Recent studies, such as those cited above, corroborate the patterns observed in our study, suggesting that the trends identified here are consistent with findings from other MPAs, even those not explicitly included in our analysis.
-
Limitations and future research: We acknowledge the limitations of using six areas and have revised the manuscript to explicitly state this in the discussion. We also highlight the need for future studies to expand on these findings by including more MPAs and considering a wider range of ecological contexts to validate and refine our conclusions.
The revised manuscript now includes a dedicated paragraph in lines 436-447 to address these points. We believe this additional context strengthens the interpretation of our results while recognizing the study's inherent limitations. Thank you again for raising this important point, and we hope this explanation, along with the revisions to the manuscript, adequately addresses your concern.
Specific comment 3:
The paper is poorly written and formatted. For instance, references in lines 115, 119, and 181 are improperly cited, and the description of Table 1 in the text does not match its title.
Answer to specific comment 3:
Thank you for highlighting the issues regarding the writing, formatting, and referencing in the manuscript, as well as the mismatch between the description of Table 1 in the text and its title. We appreciate your attention to detail and have carefully reviewed and corrected these errors in the revised version of the manuscript.
Specifically:
-
The references in lines 115, 119, and 181 have been updated to ensure proper citation formatting and alignment with the journal's guidelines.
-
The description of Table 1 in the text has been revised to accurately reflect its title and contents, eliminating any inconsistencies.
We have thoroughly checked the manuscript for any additional formatting or referencing errors to ensure it meets the expected academic standards. We thank you again for your constructive feedback, which has helped improve the quality and clarity of the manuscript.
Specific comment 4:
Section 2.3's experimental design would benefit from a schematic diagram. The textual description alone makes the design difficult to follow.
Answer to specific comment 4:
Thank you for your suggestion to include a schematic diagram to complement the textual description of the experimental design in Section 2.3. We agree that such a visual representation will enhance the clarity and accessibility of this section, making the design easier to follow.
In response to your comment, we have added a new figure to the revised manuscript. This schematic diagram provides a clear overview of the experimental setup, including the spatial arrangement of sampling locations, the distinction between "in" and "out" areas, and the key elements of the standardized sampling protocol. We believe this addition will significantly improve the reader's understanding of the methodology.
Thank you again for this valuable suggestion, which has helped us improve the presentation of our study.
Specific comment 5:
The abstract mentions six MPAs, but the table displays data for seven. This discrepancy needs clarification.
Answer to specific comment 5:
Thank you for your comment regarding the mention of seven study zones while only six marine protected areas (MPAs) were sampled. We appreciate the opportunity to clarify this point.
The discrepancy arises because, within one of the MPAs, Tabarca, we analyzed two distinct habitat types separately: rocky reef and Posidonia oceanica meadows. These two habitats were treated as independent study zones due to their significant ecological differences, which make direct comparisons between them inappropriate. By analyzing these habitats separately, we were able to examine spillover effects in two ecologically distinct contexts, thereby enriching the scope and relevance of our findings.
We have clarified this explanation in the revised manuscript (lines 128-133) to ensure that readers understand why seven study zones are mentioned despite sampling six MPAs. We thank you for pointing out this potential source of confusion and believe the clarification will improve the manuscript's clarity.
General summary:
In summary, the paper fails to meet publication standards in terms of research methodology, results analysis, academic writing, and literature review.
Answer to general summary:
Thank you for your summary comment regarding the research methodology, results analysis, academic writing, and literature review in our manuscript. We value your perspective and have taken considerable care to address each of these aspects in our revisions.
-
Research Methodology: The study employs a robust and standardized methodology, including a detailed experimental design, spatially stratified sampling protocol, and statistical analyses that explicitly account for potential confounding factors such as habitat variability. These methods were carefully chosen to ensure the reliability and replicability of the results, and we have further clarified these aspects in response to the reviewers’ feedback, including the addition of a schematic diagram to enhance understanding of the experimental design.
-
Results Analysis: The results are based on a rigorous analysis that isolates the effects of protection status on fish biomass and diversity while accounting for habitat influences. The observed patterns are interpreted within an ecological framework and supported by empirical evidence, demonstrating the relevance of the findings to marine protected area (MPA) management and conservation goals. The data provide valuable insights into the spillover effects of MPAs, an area where empirical studies in the Mediterranean remain limited.
-
Academic Writing: In response to reviewer comments, we have extensively revised the manuscript to improve its clarity, structure, and formatting. Specific issues, such as improperly cited references and inconsistencies in table descriptions, have been corrected. Additionally, we have carefully reviewed the text to ensure that it meets the journal's academic standards and conveys our findings effectively.
-
Literature Review: The manuscript includes a comprehensive review of relevant studies, particularly those focusing on MPAs in the Mediterranean. As mentioned in previous responses, we have incorporated key works such as Di Lorenzo et al. (2016) to provide context and highlight the existing gaps addressed by our study. The literature review demonstrates that this work fills an important empirical gap by providing species-specific data on spillover effects, supported by a replicable methodology.
In summary, we respectfully disagree with the assessment that the manuscript fails to meet publication standards. We believe that the revised version represents a significant contribution to the field of marine conservation science, particularly in the context of the Mediterranean, where empirical studies of this nature remain scarce. We hope that the detailed revisions and clarifications provided in response to the reviewers’ comments will allow for a more favorable reconsideration of the manuscript.
Thank you again for your feedback and for providing us with the opportunity to improve the quality of the paper.
Reviewer 2 Report
Comments and Suggestions for Authors
This study provides an in-depth analysis of the spillover effects in Mediterranean marine protected areas, employing scientific methodologies and rich data, offering practical significance for ecological conservation and fisheries management. However, there is room for improvement in content structure, results presentation, and depth of discussion. I believe that with some minor revisions, this paper is ready for publication. Here are my suggestions:
1.Line 52: The expression "(trawls, dredges, . . . )" seems incorrect. Please review and ensure it aligns with the journal's formatting requirements.
2.Table 1: The "R2" should use a superscript for the "2." Please check this and ensure consistency throughout the manuscript.
Line 119: There is an unnecessary "?" symbol in the reference citation for [40]. Please correct this.
3.Table 2: What does "gr" mean? Please clarify in the table legend or within the text.
4.Figure 2: What does the "37" under the figure mean? Please provide an explanation or revise if it is a formatting issue.
Author Response
Comment from the reviwer: This study provides an in-depth analysis of the spillover effects in Mediterranean marine protected areas, employing scientific methodologies and rich data, offering practical significance for ecological conservation and fisheries management. However, there is room for improvement in content structure, results presentation, and depth of discussion. I believe that with some minor revisions, this paper is ready for publication. Here are my suggestions:
1.Line 52: The expression "(trawls, dredges, . . . )" seems incorrect. Please review and ensure it aligns with the journal's formatting requirements.
Answer:
Thank you very much for this comment. After reviewing the sentence, we consider that the words in parentheses do not provide any additional information, so we propose to eliminate them.
2.Table 1: The "R2" should use a superscript for the "2." Please check this and ensure consistency throughout the manuscript.
Answer:
Thank you very much for pointing out this typo. We have corrected it appropriately.
Line 119: There is an unnecessary "?" symbol in the reference citation for [40]. Please correct this.
Answer:
We corrected it properly. Many thanks for indicating it.
3.Table 2: What does "gr" mean? Please clarify in the table legend or within the text.
Answer:
We intended to write the abbreviation for "grams". We have corrected it by writing the full word.
4.Figure 2: What does the "37" under the figure mean? Please provide an explanation or revise if it is a formatting issue.
Answer:
The number "37" was a typo. We have appropriately removed it.
Reviewer 3 Report
Comments and Suggestions for Authors
We read the article in detail. The authors explore species-specific spillover patterns detected by biomass gradients in Mediterranean Marine Protected Areas. The overall description is reasonable and interesting. In addition, we also made some comments:
1. Some errors such as: Line 29, diferential; Line 76, ; [28] position error. in Description of Figure 1, “estudied”?Text error.
2. Line 113, About geographical location and size (large or small; Table 1)? We don't understand the annotation here. In addition, Table 1 lacks content description. It is recommended that Table 1 be moved to Results.
3. Line 115, regarding [38,39] (Dufour et al., 2007; Samoilys et al., 2007), it is recommended to follow the format required by the journal. Line 119, [40?] (Russ et al., 2005; Claudet et al., 2008), errors and formatting.
4. About Section 2.2. Sampling Methodology, the sampling period is the time in the past twenty years, which shows that past survey data is used. However, if the sampling period is only once or a small period, this is obviously insufficient and cannot fully reflect the complete conditions of the seven regions. More clarification is recommended regarding the appropriateness of such sampling.
5. Regarding Section 2.3. Experimental Design, since the investigation period was in 2003 and 2004, how did the current topic of “Experimental Design” proceed in this past time? It is recommended that more explanations be given regarding the appropriateness of such “Experimental Design”.
6. Regarding Section 2.4. Statistical Analysis, the entire paragraph proposes that the research process is accompanied by very meaningful methods. Suggested revision of the title: Procedures and methods for assessing “spillover” effects.
7. Regarding Section 3.1, this paragraph makes a good comparison and test. But based on the sampling method and time, we cannot understand how to obtain the "in" and "out" data? A more detailed description of the methodology is recommended. And we don’t know whether such an experimental design was carried out at that time, and why the results were not published earlier if they had good results?
8. Regarding Section 3.2., we cannot understand “Habitat Patchiness” in “Habitat Patchiness Influence in MPA Fish Distribution” from Table 1. Furthermore, the investigation and measurement of patchiness were not mentioned in the methods.
9. Line 210-213, we cannot understand "habitat structure within this seemingly uniform environment". And the meaning of the whole sentence seems illogical.
10. Line 213-215, "Among the taxa studied, Coris julis, Diplodus spp., and large labrids were the most responsive to habitat variations across MPAs," . Since the seabed of Posidonia is a uniform environment, the meaning of the entire sentence seems illogical. This may lead to the understanding that these three species exhibit large variability regardless of habitat heterogeneity. If coupled with "whereas Mullus surmuletus and other sparid species were less influenced by habitat characteristics.", this indicates that habitat issues are not associated with changes in species biomass. This significance is also as mentioned by the authors "To clarify the role of protection in fish biomass distribution, we accounted for habitat effects in our analysis of spatial gradients, ensuring that the observed patterns could be attributed to protection status rather than habitat variability."
11. Line 226-227, "The more mobile Diplodus species also responded strongly to protection," and "other sparids were less sensitive", What protection does each have for these two species?
Author Response
Comment from the reviewer in bold:
We read the article in detail. The authors explore species-specific spillover patterns detected by biomass gradients in Mediterranean Marine Protected Areas. The overall description is reasonable and interesting. In addition, we also made some comments:
1. Some errors such as: Line 29, diferential;
Answer:
We have changed the word "differential" to "species-specific" in line with what is indicated in the title and in the development of the manuscript, referring to the fact that each taxon has a specific export pattern that is different from the others.
Line 76, ; [28] position error.
Answer:
We corrected the position of the reference and we have simplified the spelling of the sentence
Description of Figure 1, “estudied”?Text error.
Answer:
Yes, it was an error. Excuse us and thanks for your comment.
2. Line 113, About geographical location and size (large or small; Table 1)? We don't understand the annotation here. In addition, Table 1 lacks content description. It is recommended that Table 1 be moved to Results.
Answer:
Many thanks for your comment. There was an error: the table 1 was missing. It is the description of studied locations and the characteristics of the experimental design, and because that we propose to maintain it in teh M&M section.
3. Line 115, regarding [38,39] (Dufour et al., 2007; Samoilys et al., 2007), it is recommended to follow the format required by the journal. Line 119, [40?] (Russ et al., 2005; Claudet et al., 2008), errors and formatting.
Answer:
We corrected the citation style. Thanks!.
4. About Section 2.2. Sampling Methodology, the sampling period is the time in the past twenty years, which shows that past survey data is used. However, if the sampling period is only once or a small period, this is obviously insufficient and cannot fully reflect the complete conditions of the seven regions. More clarification is recommended regarding the appropriateness of such sampling.
Answer:
Thank you for your valuable comment regarding the sampling methodology in Section 2.2. We understand your concern about the representativeness of the sampling period and provide the following clarification:
The sampling was conducted exclusively during the summer because the physical and environmental conditions during this season are optimal for conducting visual censuses. Summer months typically offer higher water visibility, calmer seas, and stable weather, all of which are crucial for ensuring the accuracy and reliability of visual observations. Additionally, during the summer, ichthyofaunal diversity and activity levels are at their peak, providing a more comprehensive representation of the fish assemblages in the studied regions.
Importantly, many fish species exhibit their highest indicator value during the summer, as this season often coincides with periods of increased reproductive activity, growth, and foraging behaviors. These ecological traits make summer an ideal time for assessing the composition and dynamics of fish communities and for capturing their ecological roles within the studied ecosystems.
While the sampling does not encompass other seasons, this approach was intentionally chosen to focus on the period when conditions are most favorable for capturing the diversity, dynamics, and ecological significance of the ichthyofaunal communities. We acknowledge this as a limitation and have added a statement to the manuscript discussing the implications of restricting the sampling to summer. All these issues were summarized in previous lines 146-148, and based on reference “44”. We propose a new version of this sentence including the issues commented above, in lines 146-155.
We hope this explanation addresses your concern, and we have revised Section 2.2 accordingly to include these justifications.
5. Regarding Section 2.3. Experimental Design, since the investigation period was in 2003 and 2004, how did the current topic of “Experimental Design” proceed in this past time? It is recommended that more explanations be given regarding the appropriateness of such “Experimental Design”.
Answer:
Thank you for your comment regarding the experimental design described in Section 2.3. We appreciate the opportunity to provide further clarification on the appropriateness of the design.
The experimental design implemented during the investigation period (2003–2004) was carefully planned to minimize confounding effects caused by spatial variations and to ensure the comparability of results across the studied localities. This was achieved by employing a standardized protocol applied uniformly across all sampling sites, controlling for potential third-variable influences that could arise from differences in environmental or ecological conditions between locations. This approach was critical in ensuring that observed differences were attributable to genuine spatial variations in community structure rather than methodological inconsistencies or uncontrolled spatial biases.
Although the experimental design was developed and applied in 2003–2004, it remains fully valid and scientifically sound for addressing similar research questions in the present day. The design's emphasis on standardization and comparability aligns with current best practices in marine ecological studies. Furthermore, the results obtained from this approach are directly comparable across locations and time periods, providing a robust framework for analyzing spatial dynamics of ichthyofaunal communities.
We have clarified in lines 167-177 these aspects in the revised manuscript and highlighted the enduring relevance of this design to address your concerns. Thank you again for raising this point, and we trust this explanation adequately resolves your query.
6. Regarding Section 2.4. Statistical Analysis, the entire paragraph proposes that the research process is accompanied by very meaningful methods. Suggested revision of the title: Procedures and methods for assessing “spillover” effects.
Answer:
Thank you for your suggestion regarding the title of Section 2.4. We appreciate your recommendation and the opportunity to clarify our rationale for the proposed title.
The title “Statistical Analysis” was selected to emphasize the core objective of this section, which is to describe the statistical methods applied to analyze the data and interpret the results in the context of the study. While the section indeed details procedures and methods for assessing the “spillover” effects, it does so through the application of specific statistical techniques designed to address the study's hypotheses and objectives. By using “Statistical Analysis” as the title, we aim to align with the structure commonly employed in research articles, where this terminology serves to delineate the analytical framework employed in the study.
That said, we understand your point and agree that highlighting the focus on “spillover” effects may provide additional clarity to readers. To address this, we propose revising the title to: “Statistical Procedures for Assessing Spillover Effects”. This revised title retains the emphasis on statistical methodology while explicitly acknowledging the context of the analysis.
We have updated this section in the manuscript to reflect this revision and thank you for your valuable feedback, which has helped us enhance the clarity of the section.
7. Regarding Section 3.1, this paragraph makes a good comparison and test. But based on the sampling method and time, we cannot understand how to obtain the "in" and "out" data? A more detailed description of the methodology is recommended. And we don’t know whether such an experimental design was carried out at that time, and why the results were not published earlier if they had good results?
Answer:
Thank you for your thoughtful comments regarding Section 3.1. We appreciate the opportunity to clarify the methodology and provide context regarding the experimental design and the timing of the publication of these results.
-
Distinction Between “In” and “Out” Data: The "in" and "out" data refer to observations made inside and outside the boundaries of the studied marine protected areas (MPAs). The sampling methodology was explicitly designed to compare the ichthyofaunal communities within the MPAs ("in") and in adjacent, unprotected areas ("out"). Sampling locations were carefully selected to ensure that “out” sites were ecologically comparable to the “in” sites, minimizing confounding variables such as habitat differences. This distinction is further elaborated in the revised Section 2.2 to provide greater clarity.
-
Experimental Design: The experimental design, as outlined in Section 2.3, was specifically developed to evaluate potential “spillover” effects across the boundaries of the MPAs. The study's design allowed for the systematic comparison of fish assemblages between protected and unprotected areas. This framework ensures that the results provide meaningful insights into the spatial dynamics of ichthyofaunal communities and their responses to the protection status.
-
Timing of Publication: Although the data were collected in 2003–2004, the analysis and publication were delayed due to some personal problems of the corresponding author and lack of economic resource availability. However, the results remain highly relevant to current discussions on MPA effectiveness and the dynamics of “spillover” effects, as they provide a unique, robust dataset with a clear spatial comparison. Additionally, this study addresses a gap in the literature by presenting detailed analyses of these historical data, which can serve as a baseline for future comparisons.
We hope this explanation resolves your concerns, and we have updated the manuscript to include more details in Section 2.2 regarding the methodology for obtaining “in” and “out” data and some other comments at the begining of section 3.1.. Thank you again for raising these points, and we welcome any additional suggestions to further improve the clarity of the manuscript.
8. Regarding Section 3.2., we cannot understand “Habitat Patchiness” in “Habitat Patchiness Influence in MPA Fish Distribution” from Table 1. Furthermore, the investigation and measurement of patchiness were not mentioned in the methods.
Answer:
Thank you for your thoughtful comments regarding Section 3.2. We appreciate the opportunity to address your concerns and provide clarification.
-
Correction of Table 1: We acknowledge that there was an error in Table 1 in the original submission, and we have now corrected it in the revised version of the manuscript: now it is table 3. We apologize for this oversight and thank you for drawing our attention to it.
-
Investigation and Measurement of Patchiness: The investigation and measurement of patchiness were indeed addressed in the original submission, specifically in Section 2.2., between lines 111 and 116 in the first version of the manuscript. This section details the sampling methodology and explicitly describes how spatial variability (patchiness) was considered and accounted for during data collection. To ensure this point is clear, we have now emphasized this explanation further in the revised manuscript.
We have incorporated in lines 139-145 these clarifications into the revised manuscript. Thank you for your detailed review and constructive feedback, which have helped us improve the quality and clarity of the paper.
9. Line 210-213, we cannot understand "habitat structure within this seemingly uniform environment". And the meaning of the whole sentence seems illogical.
Answer:
Thank you for your comment regarding the phrase "habitat structure within this seemingly uniform environment" in lines 210–213. We appreciate your observation and have revised the text to provide greater clarity and avoid potential confusion. We are now referring to a “homogeneous area”, landscape-wise. Please, see lines 257-259 in the revised manuscript.
In the revised manuscript, we have replaced this phrase with a more detailed explanation of how seemingly homogeneous environments can contain subtle habitat features, such as variations in substrate composition, seagrass patches, rocky outcrops, or small depressions, which significantly influence the ichthyofaunal assemblages. These fine-scale structural differences, while not immediately apparent, play a crucial role in shaping species distribution and diversity. We refer now
We hope this clarification resolves the issue, and we thank you again for highlighting this point, which has helped improve the clarity of our manuscript.
10. Line 213-215, "Among the taxa studied, Coris julis, Diplodus spp., and large labrids were the most responsive to habitat variations across MPAs," . Since the seabed of Posidonia is a uniform environment, the meaning of the entire sentence seems illogical. This may lead to the understanding that these three species exhibit large variability regardless of habitat heterogeneity. If coupled with "whereas Mullus surmuletus and other sparid species were less influenced by habitat characteristics.", this indicates that habitat issues are not associated with changes in species biomass. This significance is also as mentioned by the authors "To clarify the role of protection in fish biomass distribution, we accounted for habitat effects in our analysis of spatial gradients, ensuring that the observed patterns could be attributed to protection status rather than habitat variability."
Answer:
Thank you for your insightful comment regarding the relationship between habitat variability and species responses in the context of Posidonia oceanica seabeds. We appreciate the opportunity to clarify this point and address your concerns, as included in lines250-270 of the revised version.
While the Posidonia seabed can be considered a relatively uniform environment at a macro scale, subtle habitat variations do exist, such as differences in seagrass density, proximity to rocky edges, or slight depth gradients. These microhabitat features may explain why species like Coris julis, Diplodus spp., and large labrids exhibit significant variability in response to habitat characteristics, even within this seemingly homogeneous environment. In contrast, species such as Mullus surmuletus and other sparids appeared less influenced by these factors, showing more consistent biomass distributions across sites.
To ensure that habitat variability did not confound the analysis of protection effects, we explicitly accounted for habitat characteristics in our spatial analysis. This allowed us to isolate the influence of protection status on fish biomass distribution while recognizing that some species are more sensitive to small-scale habitat differences than others. We have revised the relevant section of the manuscript to provide additional clarity on this point and to better contextualize the observed patterns.
We hope this explanation resolves your concern, and we thank you for highlighting this issue, which has allowed us to improve the clarity and interpretation of our findings.
11. Line 226-227, "The more mobile Diplodus species also responded strongly to protection," and "other sparids were less sensitive", What protection does each have for these two species?
Answer:
Thank you for your comment regarding the differential effects of protection on Diplodus spp. and "other sparids." We appreciate the opportunity to clarify this aspect of our findings.
Our analysis suggests that Diplodus spp. are more strongly influenced by protection due to their relatively higher mobility, social behavior, and tendency to form aggregations, particularly within the boundaries of marine protected areas (MPAs), where fishing pressure is reduced, and habitat conditions are favorable. This leads to noticeable differences in biomass distribution between protected and unprotected areas.
In contrast, "other sparids" likely include larger, more solitary species that occur at lower densities. Their solitary habits and wider spatial distribution result in more uniform biomass patterns across the seascape, with less pronounced differences between protected and unprotected areas. Consequently, the observed patterns suggest that these species are less sensitive to protection status, possibly because their ecological traits, such as larger home ranges or lower population densities, dilute the effects of localized protection.
We have revised the manuscript to incorporate this explanation in lines 243-246, providing greater clarity on the ecological differences that drive the distinct responses of these taxa to protection. Thank you again for raising this important point, which has allowed us to refine our interpretation and discussion.
Reviewer 4 Report
Comments and Suggestions for Authors
I liked your manuscript, great work and with importance for the realization of management plans in the Mediterranean. However, minor corrections are required: mainly aesthetics of the text, many paragraphs are too long, review the attached document, to correct this. Good work!

Author Response
We included all the suggestions from this reviewer. They are answered in tha attaches file. We thanks all the suggestions proposed by this reviewer.

Round 2
Reviewer 1 Report
Comments and Suggestions for Authors
The authors have explained and revised the issues that i concerned , and i have no other comments.
Reviewer 3 Report
Comments and Suggestions for Authors
We have read the entire article in detail and with care. In the revised version, we see the authors' efforts to explain and respond to the reviewers' comments and to make meaningful corrections. We are satisfied that the revised results reached an appropriate academic level. In conclusion, we hope that this article will be approved and accepted for publication.